# Sign2GPT: Leveraging Large Language Models for Gloss-Free Sign Language Translation

**Ryan Wong**[1], **Necati Cihan Camgoz**[2], **Richard Bowden**[1]
[1]University of Surrey, [2]Meta Reality Labs
{r.wong, r.bowden}@surrey.ac.uk, neccam@meta.com

## Abstract

Automatic Sign Language Translation requires the integration of both computer vision and natural language processing to effectively bridge the communication gap between sign and spoken languages. However, the deficiency in large-scale training data to support sign language translation means we need to leverage resources from spoken language. We introduce, Sign2GPT, a novel framework for sign language translation that utilizes large-scale pretrained vision and language models via lightweight adapters for gloss-free sign language translation. The lightweight adapters are crucial for sign language translation, due to the constraints imposed by limited dataset sizes and the computational requirements when training with long sign videos. We also propose a novel pretraining strategy that directs our encoder to learn sign representations from automatically extracted pseudo-glosses without requiring gloss order information or annotations. We evaluate our approach on two public benchmark sign language translation datasets, namely RWTH-PHOENIX-Weather 2014T and CSL-Daily, and improve on state-of-the-art gloss-free translation performance with a significant margin.

## 1 Introduction

Sign languages are the primary form of communication for millions of Deaf individuals. Sign languages make use of complex visual gestures as a form of communication (Braem & Sutton-Spence, 2001). Automatic sign language translation is a challenging task for both natural language processing and computer vision, as it requires understanding of both sign and spoken language semantics.

Many prior studies have heavily relied on gloss annotations as a means to achieve better translation performance (Camgoz et al., 2018; 2020b; Ye et al., 2023; Yao et al., 2023; Zhang et al., 2023a), where gloss annotations represent written descriptions of signs. These glosses are provided in sign order, aiding in the regularization of models to learn valuable sign features necessary for continuous sign language recognition and translation. However, creating datasets with gloss annotations is resource-intensive and time-consuming. Hence, a recent trend is to shift towards gloss-free sign language translation (Camgoz et al., 2020a; Yin et al., 2023; Zhou et al., 2023), which is the primary focus of our paper. Gloss-free sign language translation poses significant challenges due to the unique grammatical structures and vocabulary of sign languages. Limited data availability, individual signing style variation and long sign videos add to the complexity of the task.

In this paper, we propose Sign2GPT for sign language translation to address the aforementioned challenges. As depicted in Figure 1, Sign2GPT utilizes pretrained large vision (Oquab et al., 2023) and language models (Lin et al., 2021), benefiting from their extensive training on large-scale datasets to improve sign language translation performance. We also present a novel pretraining strategy before the translation task, which encompasses two key components. Firstly, we develop an algorithm designed to automatically generate pseudo-glosses. Secondly, we introduce a prototype driven method for pretraining the sign encoder using the pseudo-glosses. Notably, our approach eliminates the necessity of manual gloss annotations as well as glosses to be in sign order.

To combat over-fitting and address memory constraints resulting from long video sequences during training, we adopt a strategy of freezing the external pretrained models and utilize specialized

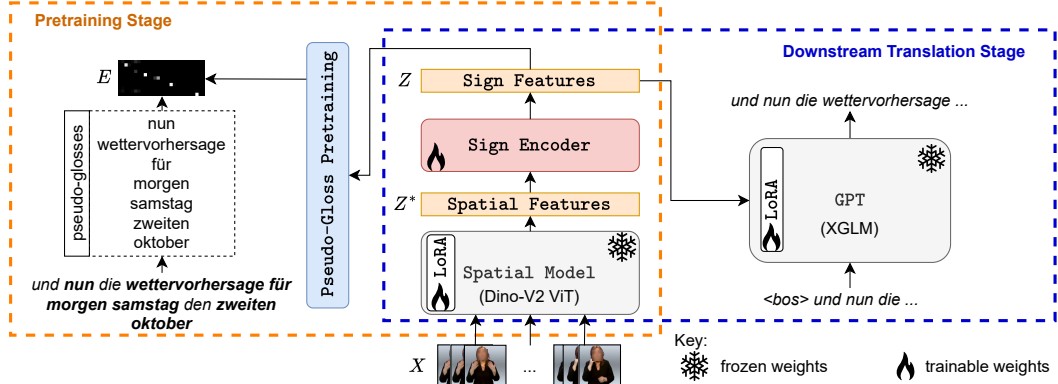

Figure 1: Overview of Sign2GPT, which consists of a pretraining stage that makes use of pseudo-glosses and downstream translation that leverages a frozen GPT model.

low-rank adapters. Inspired by Hu et al. (2021), we employ low-rank adapters as a method for facilitating the adaptation of frozen Generative Pretrained Transformers (GPT) (Radford et al., 2018; Lin et al., 2021) as well as a Vision Transformer (ViT) (Dosovitskiy et al., 2020; Oquab et al., 2023) to the specialized domain of sign language. Our contributions can be summarized as follows: (1) We introduce an end-to-end gloss-free sign language model, Sign2GPT, designed for sign language translation, leveraging a frozen GPT language model, (2) We propose a novel pseudo-gloss pretraining strategy, utilizing automatically extracted pseudo-glosses from sentences to pretrain the sign encoder. (3) Sign2GPT demonstrates significant performance improvements over previous gloss-free sign language translation approaches, offering a promising pathway for adapting frozen language and vision models to the domain of sign language translation.

## 2 RELATED WORK

Isolated Sign Recognition (ISR), the task of identifying individual signs within a sign video, has been extensively studied in the literature. Many ISR approaches leverage deep learning, adapting action recognition models (Carreira & Zisserman, 2017; Tran et al., 2018; Yan et al., 2018; Jiang et al., 2021; Albanie et al., 2020) on datasets such as WLASL (Li et al., 2020a), MSASL (Joze & Koller, 2018) and AUTSL (Sincan & Keles, 2020). New approaches have explored the integration of linguistic priors to enhance the accuracy of ISR models (Zuo et al., 2023; Wong et al., 2023). This is achieved through the utilization of pretrained word embeddings (Joulin et al., 2016) as features to regularize the models during training.

While these methodologies have been successful for ISR, they address only a portion of the sign language translation challenge. Consequently, new datasets for Continuous Sign Language Recognition have been developed, such as RWTH-PHOENIX-Weather 2014 (Koller et al., 2015) and CSL-Daily (Zhou et al., 2021). These datasets consist of sequences of coarticulated signs in a continuous video, accompanied by target labels known as glosses. These glosses are arranged in sign order and serve as an intermediary representation bridging the gap between sign videos and spoken language translation. Many approaches use Connectionist Temporal Classification (CTC) loss (Graves et al., 2006) to localize and recognize glosses within sign videos (Camgoz et al., 2018; 2020b; Zhou et al., 2021; Cui et al., 2017; Zheng et al., 2023) but this relies on the monotonicity of labels and content.

Automatic Sign Language Translation (SLT) is one of the main goals of computational sign language research, aimed at bridging the communication gap between sign languages and spoken languages. SLT datasets, such as RWTH-PHOENIX-Weather 2014T (Camgoz et al., 2018) and CSL-Daily (Zhou et al., 2021), contain video sequences of signers and corresponding spoken language sentences. These videos are often much longer than action recognition datasets, which typically use a fixed-length short sequence as input (Soomro et al., 2012; Carreira & Zisserman, 2017).

SLT is generally divided into gloss-based and gloss-free translation. Gloss-based SLT involves translating sign language into spoken language with the aid of gloss annotations via the utilization of the aforementioned Continuous Sign Language Recognition methods. The gloss annotations are manually annotated by expert annotators and depict sign type and order. Initial approaches used

CNN and RNN-based models (Camgoz et al., 2018), but transformers have gained prominence in neural machine translation (Vaswani et al., 2017), leading to the development of Transformer-based approaches for SLT (Camgoz et al., 2020b;a; Chen et al., 2022b; Gan et al.; Zhang et al., 2023a). Further improvements by progressive pretraining of large language models are also achieved by using manually annotated sign-ordered gloss as supervision (Chen et al., 2022a).

Recently, researchers have explored gloss-free sign language translation to avoid the dependencies of manually created gloss annotations. Approaches of finding glosses include using sign spotters (Sincan et al., 2023; Shi et al., 2022; Momeni et al., 2022), but these models are limited to the domain of the sign datasets and the vocabulary it was trained on. We take an alternative approach by automatically creating pseudo-glosses from the spoken language sentences. By using sentences and sign language priors, we automatically generate pseudo-glosses, overcoming limitations associated with pretrained sign spotting models and manually annotated labels. While techniques for verifying the existence of words exist (Zhao et al., 2021), they utilize separate models for each vocabulary item and two-stage translation system, incurring high training costs. In contrast, we propose a singular model that efficiently determines both the localization and existence of pseudo-glosses. This model serves as the foundation for an end-to-end solution in sign language translation.

Other methods of gloss-free SLT make use of domain transfer through the integration of linguistically pretrained models to enhance SLT performance by using pretrained encoder-decoder models like mBart (Liu et al., 2020) and T5 (Raffel et al., 2020) models to improve translation quality (Zhou et al., 2023; Uthus et al., 2023). We instead use a frozen Generative Pretrained Transformer (GPT) (Lin et al., 2021; Radford et al., 2018) as the decoder to enhance translation performance, rather than fine-tuning or progressively pretraining encoder-decoder language models used in previous approaches.

Recently Low-Rank Adapters (Hu et al., 2021), which update only a small number of weights, have been proposed to adapt frozen LLMs to spoken language tasks. They have also been applied to multi-modal learning for general visual understanding related tasks (Zhang et al., 2023c; Gao et al., 2023; Zhang et al., 2023b). These approaches use features extracted from CLIP (Radford et al., 2021) which are not suitable as a sign language representation due to being trained on image captions. Our proposed pretraining approach focuses on visual-linguistic sign language features and subsequently adapting them to spoken language models, harnessing the capabilities of LLMs.

## 3 METHOD

Our approach aims to leverage the linguistic knowledge inherent in LLMs to enhance Sign Language Translation (SLT). We address two main challenges: handling memory-intensive sign language videos with numerous frames and creating sign representations that seamlessly integrate with LLMs. In the following subsections, we outline our model architecture (Figure 1) and training strategy to tackle these challenges and enhance SLT using LLMs.

### 3.1 MODEL ARCHITECTURE

**Spatial Backbone.** Our video-based sign language translation framework relies on the spatial model, designed to extract spatial features, $Z^*$, from input video frames denoted as $X = \{x_0, x_1, ..., x_{T^*}\}$ with $T^*$ frames. These features of dimension $C$, represented as $Z^* \in \mathbb{R}^{T^* \times C}$, are subsequently used by the sign encoder. We employ the Dino-V2 Vision Transformer (Oquab et al., 2023), specifically the ViT-S/14 variant. Dino-V2 is a self-supervised vision model known for its robust feature extraction capabilities across various visual tasks, making it a suitable choice for our framework. Fine-tuning Dino-V2 is essential for adapting it to the unique characteristics of sign language translation datasets.

The spatial model often involves a significant number of parameters, posing memory and computational challenges. To address this, we adopt LoRA (Low-Rank Adapters), a lightweight adaptation technique. LoRA has been shown to be effective in finetuning LLMs (Hu et al., 2021) and Diffusion models (Roich et al., 2022; Ruiz et al., 2022; Gal et al., 2022). LoRA is applied to the top encoder layers, targeting fully connected layers in the MLP and Multi-Head Attention as shown in Figure 2 (left). We utilize the class token's output as a feature vector, undergoing linear transformation and batch normalization to produce $Z^* \in \mathbb{R}^{T^* \times C}$, feeding it into our sign encoder.

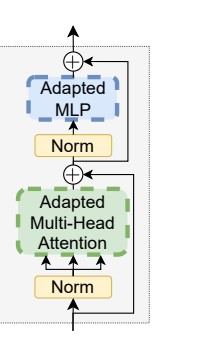 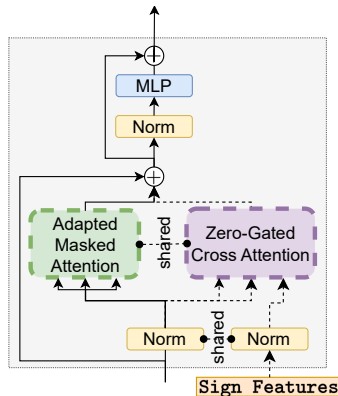

**Adapted Spatial Layer**     **Gated Cross Attention Decoder Layer**

Figure 2: Overview of adapting layers in the spatial model layers (left) and decoder layer (right). We make use of adapters that introduce new low-rank weights to blocks shown by the dashed lines while keeping the original pretrained weights frozen.

**Sign Encoder.**   We utilize spatial features, $Z^*$, as input to our sign encoder, aiming to learn spatio-temporal sign representations. Our translation model must handle sequences often comprising hundreds of frames. To meet this requirement, we designed a spatio-temporal transformer model, drawing inspiration from prior sign language translation approaches (Camgoz et al., 2020b; Yin et al., 2023). This model incorporates two crucial modifications to enhance efficiency and effectiveness.

Firstly, to address the challenge of processing many frames, we employ temporal downsampling after specific layers within our encoder. This downsampling reduces the temporal dimension from $T^*$ to $\frac{T^*}{2}$ using strided averaging with a kernel size of three and a stride of two. This design was chosen to balance between computational efficiency and the preservation of temporal information.

Secondly, we use local self-attention with a window size of seven, a technique proven to be highly effective in SLT tasks (Yin et al., 2023). This local attention mechanism is integrated with the temporal downsampling, extending the model's temporal receptive field deeper into the network. This streamlined approach not only conserves memory but also minimizes redundancy, making it highly compatible with the subsequent decoder model. The output sign representation is denoted as $Z \in \mathbb{R}^{T \times C}$, with $T$ representing the output temporal dimension which has been downsampled where $T = \frac{T^*}{2}$.

**Language Decoder.**   In the decoder, we adapt the XGLM model (Lin et al., 2021), a multilingual GPT Language Model known for its versatility in few-shot learning on text data. We chose the 1.7B parameter variant of XGLM to balance performance and memory utilization. We draw inspiration from successful language model adaptation techniques like zero-gated cross-attention and LoRA which have been used to adapt LLMs to different textual and multi-model tasks (Gao et al., 2023; Zhang et al., 2023c;b; Alayrac et al., 2022).

Before passing the sign features to the decoder, we map our sign features to the decoder's dimension with a linear layer $\mathtt{FC}_m$. To adapt the XGLM decoder for sign language translation, we employ the zero-gated multi-head cross-attention to the decoder layer as shown in Figure 2 (right). This enhancement shares weights from the pretrained masked multi-head attention and integrates a separate LoRA for masked multi-head attention (Adapted Masked Attention) and cross-attention (Zero-Gated Cross Attention). The sign features are first passed through the frozen decoder's layer normalization and then used as keys for cross-attention. To integrate sign features without overshadowing linguistic features, we employ gated scaled dot-product attention into the cross-attention:

$$\mathrm{GatedAttention}(Q, K, V) = \left( \boldsymbol{g} \times \mathtt{softmax} \left( \frac{QK^T}{\sqrt{d_k}} \right) \right) V \tag{1}$$

K and V represent the inputs from the key and value derived from the sign features, while Q originates from the textual features. $\boldsymbol{g}$ is a learnable gate parameter for each attention head which is clamped between 0 and 1 and initialized to zero to preserve linguistic knowledge at the start of training. This modification allows XGLM to adapt seamlessly to sign language. The gate parameter and

LoRA parameters drive this adaptability, gradually incorporating sign features while leveraging linguistic knowledge. As depicted in Figure 2 (right), the outputs from the Adapted Masked Attention and Zero-Gated Cross Attention are summed together.

## 3.2 TRAINING STRATEGY

Our model architecture (Figure 1) allows direct video-to-text training for SLT. We prioritize high-quality sign features by concentrating most of the trainable parameters in the sign encoder. We freeze the pretrained vision and language model and use the adapters for sign language domain transfer. This approach ensures our model's primary focus on capturing and utilizing sign language features and then leverages the language model's linguistic ability to adapt the features to spoken language translation. The result is an architecture tailored to SLT, enabling effective translation from video input to textual output. In our framework, the spatial model and decoder are pretrained from large-scale datasets while the sign encoder is randomly initialized. We therefore develop a gloss-free strategy to pretrain the sign encoder.

## 3.3 PRETRAINING STAGE

**Pseudo-gloss generation.** To perform pseudo-gloss generation, we extract pseudo-gloss from each spoken language sentence using the spaCy natural language processing library (Honnibal et al., 2020). In the case of German (Phoenix14T), this involves lemmatization, while for Chinese (CSL-Daily), it encompasses word segmentation. The preprocessing step enables the extraction of the base forms of words.

Following this step, we apply Parts-of-Speech (POS) tagging, retaining only words categorized as ["NOUN", "NUM", "ADV", "PRON", "PROPN", "ADJ", "VERB"]. This filtering process prioritizes words that are most likely to convey meaningful information, ensuring that our extracted pseudo-glosses are semantically relevant, retain most of the sentence context and have potential gloss correspondence. The lemmatized words (in German) or segmented words (in Chinese) that satisfy this filter are considered as pseudo-glosses. It's important to highlight that our pseudo-glosses are in spoken language order, unlike manually annotated glosses which are in sign order. Consequently, conventional approaches such as the CTC loss (Graves et al., 2006) are not suitable for our pseudo-glosses.

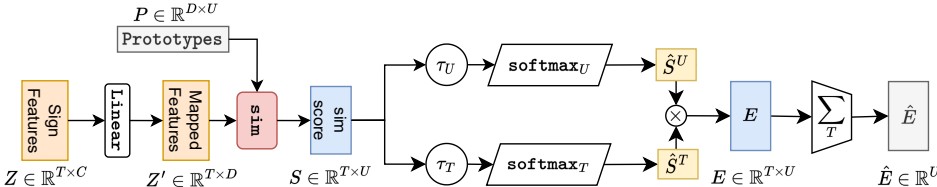

Figure 3: Overview of pretraining process, which takes the sign features as input and predicts the existence of pseudo-glosses.

**Pseudo-gloss pretraining.** Our goal is to enable the sign encoder to learn visual-linguistic representations. CLIP's ability to unify visual and linguistic domains (Radford et al., 2021) motivates the design of our sign encoder to develop sign features using pseudo-glosses for Sign Language Translation (SLT), as illustrated in Figure 3.

We generate prototypes for each pseudo-gloss. The aim is to ensure that the sign encoder generates representations that closely align with these prototypes when they are present within the sign video. These prototypes are initialized with word embeddings obtained from fastText (Joulin et al., 2016), each of dimension $D = 300$. Additionally, we include an extra prototype initialized with zeros, which serves as the prototype for sign transitions or non-sign-related components. As a result, the prototype matrix takes the form of $P \in \mathbb{R}^{D \times U}$.

The sign features $Z = \{z_0, z_1, ..., z_i, ..., z_T\}$ are learned to be aligned with the prototypes if the associated pseudo-gloss exists within the sign video. We therefore project $Z$ to $Z' \in \mathbb{R}^{T \times D}$ through a linear layer and compute the cosine similarity between the prototypes and each of the projected

sign features such that:

$$s_i = \texttt{sim}(z_i', P) = \frac{z_i' \cdot P}{\|\hat{z}_i\|\|P\|} \tag{2}$$

which is the cosine similarity score for the $i^{th}$ feature index in $T$ and $S = \{s_0, s_1, ..., s_T\} \in \mathbb{R}^{T \times U}$, $-1 \leq s_i \leq 1$. High scores indicate similarity, and low scores indicate dissimilarity. We then introduce temporal probability ($\hat{S}^T$) and prototype probability ($\hat{S}^U$) scores using temperature-scaled softmax operations across the time and prototype axis:

$$\hat{S}^T = \texttt{softmax}_T(S/\tau_T) \tag{3}$$

$$\hat{S}^U = \texttt{softmax}_U(S/\tau_U) \tag{4}$$

These scores emphasize temporal and class-related aspects of similarity and ensure they fall within the range of 0 to 1. Learnable scaling factors, $\tau_T$ and $\tau_U$, modulate the extent of temporal and class similarity, allowing control of their influence on prototype creation. Element-wise multiplication of $\hat{S}^T$ and $\hat{S}^U$ yields $E \in \mathbb{R}^{T \times U}$, forming the basis for prototype localization. To discern the presence or absence of prototypes, we aggregate $E$ values over the temporal dimension, resulting in $\hat{E} \in \mathbb{R}^U$ such that:

$$\hat{E}_j = \sum_{i=0}^{T} E_{i,j} = \sum_{i=0}^{T} (\hat{S}_{i,j}^T \times \hat{S}_{i,j}^U) \tag{5}$$

where $0 \leq E_{i,j} \leq 1$. High $\hat{E}_j$ value for the $j^{th}$ prototype indicates the presence of the prototype, while low values signify the absence. We employ binary cross-entropy loss to train the model, optimizing the presence or absence of prototypes. For our case each prototype is assigned as 1 if the pseudo-gloss exists within the sign video and 0 if not.

Note that during pretraining the learned sign representations are temporally invariant, therefore we also add sinusoidal positional encoding before $\texttt{FC}_m$ for the translation task. The resulting pretraining allows the weights to be initialized with vision priors (spatial model), sign priors (sign encoder) and linguistic priors (decoder). This enables us to then use these models for the downstream translation task.

## 4 RESULTS

### 4.1 DATASETS AND EVALUATION PROTOCOL

We evaluate our approach using two distinct sign language translation datasets. **RWTH-PHOENIX-WEATHER-2014T (Phoenix14T)** (Camgoz et al., 2018) is a German Sign Language dataset for sign to spoken language translation tasks. It encompasses translations from German Sign Language to the German language, derived from German weather broadcast videos. **CSL-Daily** (Zhou et al., 2021) is a translation dataset focusing on Chinese Sign Language to Chinese. It comprises of lab recorded videos that cover various daily interaction topics, including travel, family life, bank service and shopping.

We evaluate the translation performance using standard metrics commonly employed in sign language translation. These metrics include BLEU (Bilingual Evaluation Understudy) (Papineni et al., 2002) and ROUGE-L (Lin, 2004) scores. BLEU measures the similarity between machine-generated and reference translations based on n-gram overlap. ROUGE-L assesses translation quality by calculating the length of the longest common sub-sequence between generated and reference texts.

### 4.2 TRAINING SETTINGS

The model is trained end-to-end with a batch size of 8 on two A100 GPUs, subsampling every second frame. Due to memory constraints we apply spatial adapters to the top 3 layers of the spatial model. The sign encoder is a 4 layer transformer with hidden dimension of 512, 8 attention heads and intermediate size of 2048. The temporal downsampling is applied after the 2nd layer. Training is conducted in bfloat16 and Flash attention v2 (Dao, 2023) is used to optimise memory usage. We explain further details of the libraries and models in Appendix A.1. We employ the Adam optimizer (Kingma & Ba, 2014) with a learning rate of $3 \times 10^{-4}$ and weight decay of 0.001. Training spans

100 epochs with gradient clipping of 1.0 and includes a one-cycle cosine learning rate scheduler (Loshchilov & Hutter, 2016) with warmup for the initial 5 epochs. Data augmentation techniques, such as color jitter, random resized cropping from $256 \times 256$ to $224 \times 224$ pixels, frame rotations and horizontal flips are consistently applied across all frames within video sequences. During evaluation, we use center cropping to $224 \times 224$ pixels.

**Pretraining.** We initialize the prototype ($\tau_U$) and time temperature ($\tau_T$) to 0.1. In Table 1(a), we present the counts of extracted pseudo-glosses for each dataset using the approach described in Section 3.3. For the CSL-Daily dataset, the number of pseudo-glosses significantly exceeds the vocabulary size. This discrepancy arises because the vocabulary is based on Chinese characters, while pseudo-glosses represent word segmentation that convey meaning.

**Downstream Translation.** We utilize cross-entropy loss with label smoothing set to 0.1 during training. The LoRA rank and alpha values are both set to 4. During inference, we employ a beam search with a width of 4. In Table 1(b), we present detailed information regarding the number of trainable parameters for each component of our network during the downstream training phase. Notably, the sign encoder contains a significantly larger number of trainable parameters, emphasizing its role in learning sign features. For the CSL-Daily dataset, we implement additional tokenization processing to handle unknown tokens, as elaborated in Appendix A.2.

Table 1: Training setting with (a) the number of pseudo-glosses during pretraining and (b) parameter counts during downstream translation.

| Dataset | # Vocab | # p-glosses |
|---|---|---|
| Phoenix14T | 2,887 | 2,533 |
| CSL-Daily | 2,343 | 7,918 |

(a) Vocabulary versus pseudo-glosses per dataset

| Component | # Params | # Trainable |
|---|---|---|
| Spatial | 22,328,448 | 271,872 |
| Sign Encoder | 12,613,632 | 12,613,632 |
| Decoder | 1,736,710,528 | 3,803,520 |
| **Total** | 1,771,652,608 | 16,689,024 |

(b) Number of model parameters during translation

### 4.3 COMPARISONS WITH STATE-OF-THE-ART METHODS

**Results on Phoenix14T.** In Table 2, we present a comparative analysis between our approach and state-of-the-art methods for sign language translation on Phoenix14T. We observe an approximate improvement of 1.1 BLEU4 on the test set when doing pseudo-gloss pretraining (*PGP*) followed by downstream translation (*Sign2GPT(w/PGP)*). Our gloss-free approach demonstrates substantial improvements in BLEU-1,2,3 scores, surpassing the performance of previous gloss-free methods and reduces the performance gap between gloss-free and gloss-based SLT. This progress can be attributed to our approach's capability to learn representations for individual pseudo-gloss, in contrast to prior gloss-free methods that primarily focused on sentence-based representations. Notably, even without pretraining (*Sign2GPT*), our approach remains competitive with previous state-of-the-art (SOTA) gloss-free results. We also find that downstream training with only the trainable decoder parameters and extract sign features from our frozen pretrained stage model, *Sign(Z)2GPT(w/PGP)*, is able to achieve competitive results with under 4 million parameters showing the effectiveness of our learnt sign representations.

**Results on CSL-Daily.** In Table 3, we present the results of our SLT experiments on the CSL-Daily dataset. We observe a substantial performance increase in BLEU-4 compared to previous SOTA gloss-free results, with an approximate improvement of 4.4 BLEU-4 when utilizing our pretraining (*Sign2GPT(w/PGP)*). *Sign2GPT(w/PGP)* significantly outperforms training without PGP (*Sign2GPT*), with a notable 3.5 BLEU-4 improvement on the CSL-Daily dataset.

### 4.4 QUALITATIVE RESULTS

We visually demonstrate the effectiveness of our pseudo-gloss pretraining phase for sign localization in Figure 4 and Appendix A.4. To achieve this, we leverage the output values of $E$, with dimensions

Table 2: Comparison of test set results on Phoenix14T. We present our gloss-free results for three experimental settings: (1) Without pseudo-gloss pretraining (**Sign2GPT**), (2) with pseudo-gloss pretraining (**Sign2GPT(w/PGP)**), and (3) extracted features $Z$ from the frozen spatial and sign encoder model that has been trained with pseudo-gloss pretraining (**Sign($Z$)2GPT(w/PGP)**).

| Method | Test Set | | | | |
|---|---|---|---|---|---|
| | **BLEU1** | **BLEU2** | **BLEU3** | **BLEU4** | **ROUGE** |
| **Gloss-based** | | | | | |
| SL-Transformer (Camgoz et al., 2020b) | 46.61 | 33.73 | 26.19 | 21.32 | – |
| BN-TIN-Transf.+BT (Zhou et al., 2021) | 50.80 | 37.75 | 29.72 | 24.32 | 49.54 |
| MMTLB (Chen et al., 2022a) | 53.97 | 41.75 | 33.84 | 28.39 | 52.65 |
| SLTU$_{\text{NET}}$ (Zhang et al., 2023a) | 52.92 | 41.76 | 33.99 | 28.47 | 52.11 |
| TwoStream-SLT (Chen et al., 2022b) | 54.90 | 42.43 | 34.46 | 28.95 | 53.48 |
| **Gloss-free** | | | | | |
| NSLT (Camgoz et al., 2018) | 29.86 | 17.52 | 11.96 | 9.00 | 30.70 |
| TSPNet (Li et al., 2020b) | 36.10 | 23.12 | 16.88 | 13.41 | 34.96 |
| CSGCR (Zhao et al., 2021) | 36.71 | 25.40 | 18.86 | 15.18 | 38.85 |
| GASLT (Yin et al., 2023) | 39.07 | 26.74 | 21.86 | 15.74 | 39.86 |
| GFSLT (Zhou et al., 2023) | 41.39 | 31.00 | 24.20 | 19.66 | 40.93 |
| GFSLT-VLP (Zhou et al., 2023) | 43.71 | 33.18 | 26.11 | 21.44 | 42.49 |
| **Sign2GPT** | 45.43 | 32.03 | 24.23 | 19.42 | 45.23 |
| **Sign2GPT(w/PGP)** | **49.54** | **35.96** | **28.83** | **22.52** | **48.90** |
| **Sign($Z$)2GPT(w/PGP)** | 47.06 | 33.61 | 25.85 | 20.93 | 47.11 |

Table 3: Comparison of test set results on the CSL-Daily. We present gloss-free results for three experimental settings: (1) Without pseudo-gloss pretraining (**Sign2GPT**), (2) with pseudo-gloss pretraining (**Sign2GPT(w/PGP)**), and (3) extracted features $Z$ from the frozen spatial and sign encoder model that has been trained with pseudo-gloss pretraining (**Sign($Z$)2GPT(w/PGP)**).

| Method | Test Set | | | | |
|---|---|---|---|---|---|
| | **BLEU1** | **BLEU2** | **BLEU3** | **BLEU4** | **ROUGE** |
| **Gloss-based** | | | | | |
| SL-Transformer (Camgoz et al., 2020b) | 37.38 | 24.36 | 16.55 | 11.79 | 36.74 |
| BN-TIN-Transf.+BT (Zhou et al., 2021) | 51.42 | 37.26 | 27.76 | 21.34 | 49.31 |
| MMTLB (Chen et al., 2022a) | 53.31 | 40.41 | 30.87 | 23.92 | 53.25 |
| SLTU$_{\text{NET}}$ (Zhang et al., 2023a) | 54.98 | 41.44 | 31.84 | 25.01 | 54.08 |
| TwoStream-SLT (Chen et al., 2022b) | 55.44 | 42.59 | 32.87 | 25.79 | 55.72 |
| **Gloss-free** | | | | | |
| GASLT (Yin et al., 2023) | 19.90 | 9.94 | 5.98 | 4.07 | 20.35 |
| NSLT (Camgoz et al., 2018) | 34.16 | 19.57 | 11.84 | 7.56 | 34.54 |
| GFSLT (Zhou et al., 2023) | 37.69 | 23.28 | 14.93 | 9.88 | 35.16 |
| GFSLT-VLP (Zhou et al., 2023) | 39.37 | 24.93 | 16.26 | 11.00 | 36.44 |
| **Sign2GPT** | 34.80 | 24.00 | 17.27 | 12.96 | 41.12 |
| **Sign2GPT(w/PGP)** | **41.75** | **28.73** | **20.60** | **15.40** | **42.36** |
| **Sign($Z$)2GPT(w/PGP)** | 32.73 | 20.52 | 13.75 | 9.73 | 33.39 |

$T \times U$. The values in this matrix vary between 0 (indicating the absence of a sign) and 1 (indicating the presence of a sign), reflecting the temporal occurrence of pseudo-glosses. Our observations suggest that our pretraining approach holds promise for sign spotting, even in scenarios where precise localization is not available. This aspect may warrant further exploration as a potential avenue for future research.

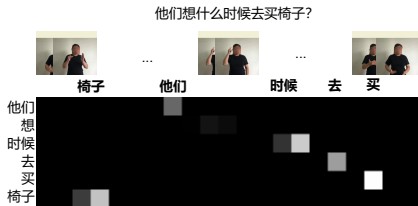

Figure 4: Visualizations of the localization capabilities of our pretraining stage. We visualize only the pseudo-glosses from the target sentence (y-axis) over time (x-axis), with whiter regions indicating a higher probability of the pseudo-gloss occurring during the time segment. We also display the localized gloss (under the video frames) based on a threshold of 0.2 on $E$.

## 4.5 ABLATION STUDY

We conduct our ablation studies on the Phoenix14T dataset, evaluating the BLEU-4 score on the development set. In Table 4, we present the results of architectural modifications made to our network. For our initial study, we conducted experiments without pretraining Sign2GPT. We observed significant performance improvements when incorporating spatial adapter. Conversely, the use of global attention led to a reduction in model performance compared to local attention. Furthermore, we investigated the impact of downsampling on the model's performance. While downsampling showed only marginal improvements in terms of the BLEU-4 score, it notably reduced computational complexity. This reduction is due to only half the number of features being passed through the decoder during training.

Our pretraining generates representations for pseudo-glosses, which inherently removes temporal information from the output sign representations. Subsequently, we investigated methods to reintroduce temporal information into the model using various approaches as described in Section 3.3. We explored learnable approaches for positional embeddings with both zero and random initialization. However, these approaches yielded minimal to no discernible benefits. In contrast, the utilization of sinusoidal positional embeddings demonstrated substantial performance improvements.

Table 4: Ablation of results on the Phoenix14T dataset showing different architecture changes with no pseudo-gloss pretraining (**Sign2GPT**) and with pseudo-gloss pretraining (**Sign2GPT(w/PGP)**).

| Architecture | BLEU4 |
|---|---|
| **Sign2GPT** | |
| Spatial Adapters + Local Attention + Downsampling | **19.55** |
| · No Spatial Adapters | 16.38 |
| · No Local Attention (+ Global Attention) | 18.56 |
| · No Downsampling on Sign Encoder | 19.30 |
| **Sign2GPT(w/PGP)** | |
| · No positional | 21.68 |
| · Learnable positional (zero init) | 21.89 |
| · Learnable positional (random init) | 21.16 |
| · Sinusoidal positional | **23.20** |

## 5 CONCLUSIONS

In this paper, we have presented a novel approach to address the challenging problem of Sign Translation in a gloss-free setting. Our method, Sign2GPT, demonstrates significant performance improvements over existing state-of-the-art techniques on the Phoenix14T and CSL-Daily datasets. We introduce a novel pretraining strategy that learns from pseudo-glosses which are generated automatically to learn word-level sign features, thereby allowing our sign encoder to be effectively pretrained without the use of manually annotated glosses. Moreover, the proposed Sign2GPT architecture presents a promising direction for the exploration of fusing visual features to spoken language models for sign language recognition and translation tasks.

## REPRODUCIBILITY STATEMENT

To facilitate reproducibility, we have provided details of the training settings in Section 4.2 with additional details of the libraries we used for the pretrained models in Appendix A.1. We also give further details of the CSL tokenization and post processing to address issues with unknown tokens in Appendix A.2.

## ACKNOWLEDGEMENTS

This work was supported by the EPSRC project ExTOL (EP/R03298X/1), SNSF project 'SMILE II' (CRSII5 193686), European Union's Horizon2020 programme ('EASIER' grant agreement 101016982) and the Innosuisse IICT Flagship (PFFS-21-47). This work reflects only the authors view and the Commission is not responsible for any use that may be made of the information it contains. Neither Necati Cihan Camgoz nor Meta were involved in the model training, evaluation, or use of the datasets.

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

## A APPENDIX

### A.1 LIBRARIES

**Spatial Backbone.** The spatial backbone is ViT-S/14 distilled with the pretrained model downloaded from `github.com/facebookresearch/dinov2`.

**Decoder.** We use the HuggingFace library for the pretrained XGLM transformer model and tokenization, specifically the pretrained weights from `facebook/xglm-1.7B`.

**Pseudo-gloss generation.** For pseudo-gloss generation we use the SpaCy library. We use the German pipeline loaded from `de_core_news_sm` for generating our pseudo-glosses for the Phoenix14T dataset. For the CSL-Daily dataset we use `zh_core_web_sm`.

**FastText embeddings.** We load the fastText embeddings from `cc.de.300.bin` for the German Phoenix14T dataset and `cc.zh.300.bin` for the Chinese CSL Dataset.

## A.2 CSL Tokenization and post processing

By using the pretrained tokenization from XGLM, we encountered unknown tokens during CSL-daily training when Chinese characters were absent from the vocabulary. To address this, we performed vocabulary expansion, utilizing the average embedding technique by Hewitt (2021) for initializing new tokens with slight noise. We maintained the consistency of our proposed architecture by keeping these new embeddings frozen throughout training, avoiding the introduction of additional learnable parameters. In total, there are 252 new tokens added to the vocabulary. We also apply post-processing of replacing punctuation tokens at inference time as the pretrained tokenizer converts punctuation to the Unicode equivalent such that '?', '!', ':' and ',' are replaced by '? ', '! ', ': ' and ', ' respectively.

## A.3 Qualitative Translation Examples

In Table 5 and 6 we present randomly selected qualitative translation results on Phoenix14T and CSL-Daily respectively. We also display the extracted pseudo-glosses for each of the target sentences which capture the core content of the sentences. Our findings demonstrate that our translation model successfully generates sentences that effectively capture the semantic content of spoken language sentences, albeit with variations in sentence structure.

Table 5: Examples of translation results on the Phoenix14T dataset.

| | |
|---|---|
| Hypothesis: | im übrigen land scheint häufig die sonne und es gibt nur wenig schauer . (**in the rest of the country the sun often shines and there are only a few showers .**) |
| Pseudo-glosses: | übrig, gebiet, sonne, nur, locker, wolke |
| Reference: | in den übrigen gebieten viel sonne und nur ein paar lockere wolken . (**lots of sun in the remaining areas and only a few loose clouds .**) |
| Hypothesis: | am tag zwölf grad an der ostsee und bis zu zwanzig grad im süden . (**twelve degrees a day on the baltic sea and up to twenty degrees in the south .**) |
| Pseudo-glosses: | tag, zwölf, grad, ostsee, zwanzig, grad, niederbayer |
| Reference: | am tag zwölf grad an der ostsee und bis zwanzig grad in niederbayern . (**on the day twelve degrees on the baltic sea and up to twenty degrees in lower bavaria .** ) |
| Hypothesis: | ich wünsche ihnen noch einen schönen abend und machen sie es gut . (**i wish you a nice evening and do well .**) |
| Pseudo-glosses: | ich, wünschen, ihnen, schön, abend, machen, sie, es, gut |
| Reference: | ich wünsche ihnen einen schönen abend und machen sie es gut . ( **i wish you a nice evening and do well .**) |
| Hypothesis: | der wind weht schwach bis mäßig aus süd bis südost . ( **the wind blows weakly to moderately from the south to southeast .**) |
| Pseudo-glosses: | dazu, wehen, schwach, wind, südost, süd |
| Reference: | dazu weht ein schwacher bis mäßiger wind aus südost bis süd . ( **in addition a weak to moderate wind blows from the southeast to the south .**) |

## A.4 Qualitative Pseudo-gloss Localization Examples

We visualize the localization capabilities of our pretraining on Phoenix14T in Figure 5 and CSL-Daily in Figure 6. We notice that our pretraining has automatic localization capabilities irrespective of the order of the pseudo-gloss when using the output $E$. This has the potential future avenues for automatically creating sign-ordered glosses based on thresholds of $E$. We observe that while we provide no information about the localization of the pseudo-glosses, our approach is able to identify the potential glosses and the sign order of these identified glosses.

Table 6: Examples of translation results on the CSL-Daily dataset.

| | |
|---|---|
| Hypothesis: | 这个地方不离饭店，走几步就到饭店的门口。**(This place is not far from the hotel, just a few steps to the door of the hotel.)** |
| Pseudo-glosses: | 可以/ 这里/ 不/ 远/ 有/ 饭馆/ 走/ 几/ 分钟/ 就/ 到 |
| Reference: | 可以，离这里不远有一个饭馆，走几分钟就到了。 **(Okay, there is a restaurant not far from here, it can be reached in a few minutes' walk.)** |
| Hypothesis: | 公司很远，他为什么不打车呢？ **(The company is far away, why doesn't he take a taxi?)** |
| Pseudo-glosses: | 公司/离家/ 很/ 远/ 他/ 为什么/ 不/ 打车 |
| Reference: | 公司离家很远，他为什么不打车？ **(The company is far from home, why doesn't he take a taxi?)** |
| Hypothesis: | 我不去爬山，我有事情要去做。 **(I'm not going to climb mountains, I have things to do.)** |
| Pseudo-glosses: | 我/ 不/ 去/ 爬山/ 我/ 有事 |
| Reference: | 我不去爬山，我有事。 **( I'm not going to climb the mountain, I have something to do.)** |
| Hypothesis: | 我喜欢下雪。 **(I like snow.)** |
| Pseudo-glosses: | 我/ 喜欢/ 冬天/ 下雪/ 太/ 美 |
| Reference: | 我喜欢冬天，下雪太美了。 **( I like winter, the snow is so beautiful.)** |

# B ADDITIONAL ABLATION STUDIES

## B.1 IMPACT OF SPATIAL BACKBONE

In Table 7, we present comparisons between the performance of a ResNet18 spatial backbone and the proposed adapted ViT model within our Sign2GPT architecture. We find that the frozen ViT model trained with DinoV2 pretraining, incorporating adapters, exhibits slightly better performance compared to the ResNet18 backbone. A noteworthy advantage of the adapted ViT model lies in the application of LoRA instead of fine-tuning all parameters. Consequently, the proposed backbone has under three hundred thousand trainable parameters, a significant reduction compared to the 11 million trainable parameters of the ResNet18 model.

Table 7: Ablation of different spatial backbones on CSL-Daily using our Sign2GPT architecture without Pseudo-Gloss Pretraining.

| Spatial Backbone | Test Set | |
|---|---|---|
| | BLEU4 | ROUGE |
| ResNet18 | 12.28 | 38.31 |
| DinoV2 (ViT-S/14) | **12.96** | **41.12** |

## B.2 PRETRAINING PERFORMANCE

In Table 8, we demonstrate the performance of our pseudo-gloss pretraining by measuring the precision, recall and F1-score for Phoenix14T and CSL-Daily using a threshold of 0.2.

Table 8: Quantitative results of pseudo-gloss pretraining on the sign language datasets.

| Dataset | Precision | Recall | F1-Score |
|---|---|---|---|
| Phoenix14T | 0.52 | 0.39 | 0.44 |
| CSL-Daily | 0.38 | 0.34 | 0.36 |

**Impact of pseudo-gloss selection.** In Table 9, we illustrate the impact of utilizing POS on precision, recall, and F1-score. Notably, when all words are used as tokens, recall significantly decreases from 0.39 to 0.28. This result validates our assertion that not all words have corresponding signs in sign language.

Table 9: Ablation of pretraining results on Phoenix14T using all words as tokens vs the selected pseudo-glosses with a threshold of 0.2

| Tokens | Precision | Recall | F1-Score |
|---|---|---|---|
| all words | **0.55** | 0.28 | 0.37 |
| pseudo-glosses | 0.52 | **0.39** | **0.44** |

**Impact of Sign Encoder.** In Table 10, we demonstrate the importance of the sign encoder on pretraining. The role of the sign encoder forms an important part of the model as it learns temporal features.

Table 10: Ablation of pretraining results on Phoenix14T with no sign encoder and the inclusion of a sign encoder.

| | Precision | Recall | F1-Score |
|---|---|---|---|
| no sign encoder | 0.50 | 0.25 | 0.33 |
| sign encoder | **0.52** | **0.39** | **0.44** |

## B.3 IMPACT OF MODEL SIZE.

In Table 11, we demonstrate the results of our approach with the smaller language model XGLM-564M. The table shows a marginal performance reduction on the Phoenix14T dataset compared to the larger XGLM-1.7B while still outperforming GFSLT-VLP (Zhou et al., 2023) with a similar size language model, mBART (Liu et al., 2020).

Table 11: Ablation of XGLM backbones on Phoenix14T using our Sign2GPT architecture.

| Backbone | Test Set | |
|---|---|---|
| | BLEU4 | ROUGE |
| GFSLT-VLP (mBART) (Zhou et al., 2023) | 21.44 | 42.49 |
| Sign2GPT(w/ PGP) (XGLM-564M) | 22.29 | 48.21 |
| Sign2GPT(w/ PGP) (XGLM-1.7B) | **22.52** | **48.90** |

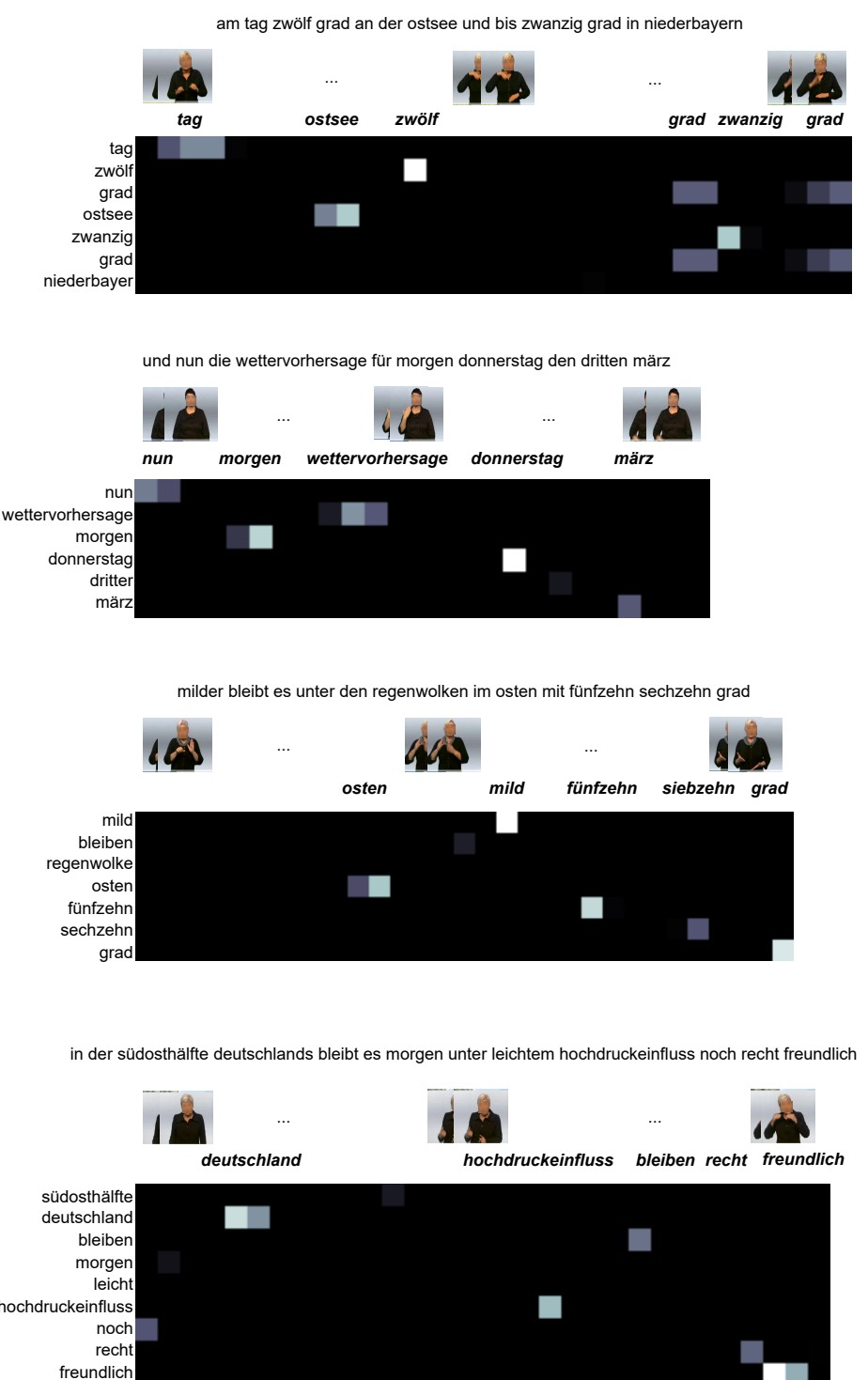

Figure 5: Visualizations of the localization capabilities of our pretraining stage on the pseudo-glosses from the Phoenix14T dataset. We visualize only the pseudo-glosses from the target sentence (y-axis) over time (x-axis), with whiter regions indicating a higher probability of the pseudo-gloss occurring during the time segment. We also display the localized gloss (under the video frames) based on a threshold of 0.2 on $E$.

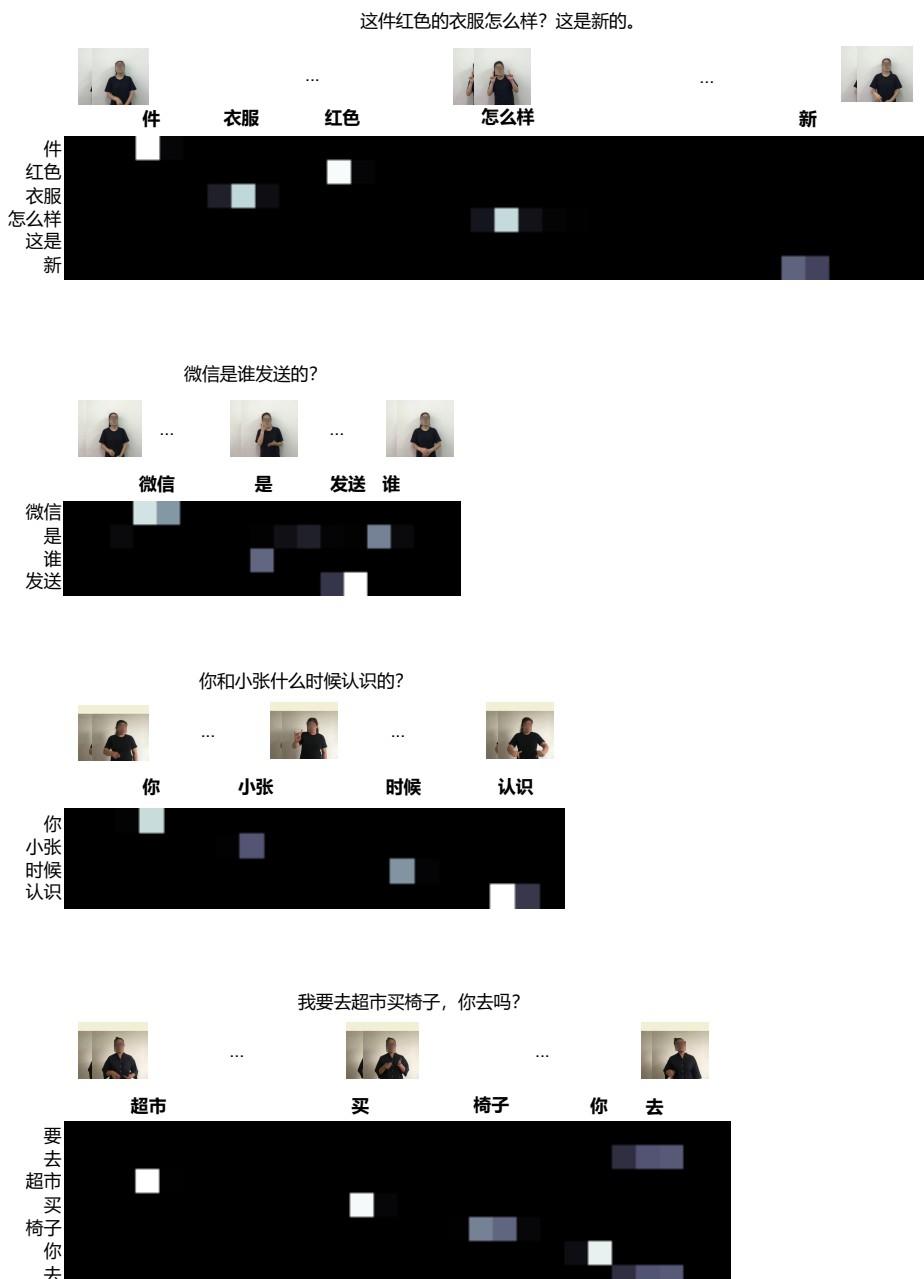

Figure 6: Visualizations of the localization capabilities of our pretraining stage on the pseudo-glosses from the CSL-Daily dataset. We visualize only the pseudo-glosses from the target sentence (y-axis) over time (x-axis), with whiter regions indicating a higher probability of the pseudo-gloss occurring during the time segment. We also display the localized gloss (under the video frames) based on a threshold of 0.2 on $E$.

