# OpenReview forum: "Sign2GPT: Leveraging Large Language Models for Gloss-Free Sign Language Translation"
_ICLR.cc/2024/Conference — ICLR 2024 poster_

### Official Review · Reviewer_hMSD · 2023-10-28

**Soundness:** 3 good
**Presentation:** 2 fair
**Contribution:** 3 good
**Rating:** 6
**Confidence:** 5

**Summary:**

The paper propose Sign2GPT, which enjoys the benefit of large pretrained language model to promote gloss-free sign language translation. The authors also propose a CLIP-styple pseudo-gloss pretraining technique to better learn visual-lingusitic representations. The overall method achieves SOTA performance on two widely adopted benchmarks, Phoenix-2014T and CSL-Daily.

**Strengths:**

1. The idea is sound. It is good to see that current large language model can be helpful for sign language understanding.
2. The proposed pseudo-gloss pretraining is novel, which can inspire future works.
3. SOTA gloss-free sign language translation performance on two benchmarks.

**Weaknesses:**

1. Using pretrained language model to boost sign language translation (SLT) is not surprisingly novel. Several works [1,2] have already verified that pretrained language model can boost (gloss-free) SLT.

2. The name of pseudo-gloss pretraining is a bit confusing, although the method itself is sound. Because the pseudo-glosses are in spoken language order, it is not quite appropriate to call them "glosses".

3. The notations of "P", "F", "D" in Table 2 and 3 are also confusing. For example, D denotes without pretraining and P denotes pretraining, then what does "P+D" mean? I suggest authors adding a "check mark" column to show which parts are pretrained.

4. I cannot find an ablation study on the pseudo-gloss pretraining. What is the performance if removing it?

5. In Table 4, removing sinusoidal positional encoding leads to the best performance. Then what poistional encoding is used by default?

6. The paper uses a new spatial backbone which is under-explored in previous sign language papers. The authors need to better motivate it. For example, is it better than other vision transformers, or is it better than widely adopted 2D/3D CNNs?

7. In figure 2, how to fuse the outputs of adapted masked attention (solid lines) and zero-gated cross attention (dashed lines)? Besides, what is adapted masked attention? I didn't see a clear definition of it.

[1] A Simple Multi-Modality Transfer Learning Baseline for Sign Language Translation, CVPR 2022

[2] Gloss-free Sign Language Translation: Improving from Visual-Language Pretraining, ICCV 2023

**Questions:**

See weakness.

---

> ### Author Response · Authors · 2023-11-16
> **Response to Reviewer hMSD (1/2)**
>
> Thanks for the review, and your insightful comments.
>
> **W1: Using pretrained language model to boost sign language translation (SLT) is not surprisingly novel. Several works [1,2] have already verified that pretrained language model can boost (gloss-free) SLT.**
>
> While we acknowledge previous research on leveraging pretrained language models for sign language translation, our approach offers several advantages over existing methods.
>
> Chen et al. [1] primarily focus on utilizing manually annotated sign-ordered gloss supervision, relying on Gloss2Text for progressive pretraining on the encoder-decoder language model. In contrast, our approach does not rely on manually annotated glosses. Instead, we concentrate on developing a method for gloss-free sign representation, utilizing a decoder-only language model without the need for progressive pretraining.
>
> Similarly, Zhou et al. [2] have explored gloss-free pretraining, but their approach involves updating the visual encoder and text decoder simultaneously. In contrast, our pseudo-gloss pretraining is independent of the language model. We keep the pretrained language model frozen by leveraging adapters, allowing the use of large-scale pretrained language models. Our architectural design outperforms their approach, as demonstrated in our results.
>
> **W2: The name of pseudo-gloss pretraining is a bit confusing, although the method itself is sound. Because the pseudo-glosses are in spoken language order, it is not quite appropriate to call them "glosses".**
>
> We termed them pseudo-glosses as they are potential glosses that are signed. However, while we extract them from the spoken language sentence our proposed pretraining is invariant to the order of the pseudo-glosses. As demonstrated in Figures 4, 5, and 6 the pretraining is also able to localize the pseudo-glosses and thus provide them in sign order.
> We are happy to change them to something more appropriate and are open to suggestions.
>
> **W3: The notations of "P", "F", "D" in Table 2 and 3 are also confusing. For example, D denotes without pretraining and P denotes pretraining, then what does "P+D" mean? I suggest authors adding a "check mark" column to show which parts are pretrained.**
>
> We have updated our tables shown below. In the table Sign2GPT without Pseudo-Gloss Pretraining is denoted as Sign2GPT and Sign2GPT(w/ PGP) indicates with Pseudo-Gloss Pretraining (PGP). This updated table clearly shows that our approach outperforms the previous state-of-the-art results. We also included Sign($Z$)2GPT(w/PGP) indicating we are giving the extracted features $Z$ to GPT, which consists of a frozen spatial and sign encoder model that has been trained with PGP, demonstrating the strength of our PGP as feature extractors for sign translation.
>
> *Translation Results on Phoenix14T:*
> Approach |BLEU1|BLEU2|BLEU3|BLEU4|ROUGE|
> |-|-|-|-|-|-|
> NSLT|29.86|17.52|11.96|9.00|30.70|
> TSPNet|36.10|23.12|16.88|13.41|34.96|
> CSGCR|36.71|25.40|18.86|15.18|38.85|
> GASLT|39.07|26.74|21.86|15.74|39.86|
> GFSLT|41.39|31.00|24.20|19.66|40.93|
> GFSLT-VLP|43.71|33.18|26.11|21.44|42.49|
> |||||||
> **Sign2GPT**|45.43|32.03|24.23|19.42|45.23|
> **Sign2GPT(w/ PGP)**|**49.54**|**35.96**|**28.83**|**22.52**|**48.90**|
> |||||||
> **Sign($Z$)2GPT(w/PGP)**|47.06|33.61|25.85|20.93|47.11|
>
> *Translation Results on CSL-Daily:*
> Approach |BLEU1|BLEU2|BLEU3|BLEU4|ROUGE|
> |-|-|-|-|-|-|
> GASLT|19.90|9.94|5.98|4.07|20.35|
> NSLT|34.16|19.57|11.84|7.56|34.54|
> GFSLT|37.69|23.28|14.93|9.88|35.16|
> GFSLT-VLP|39.37|24.93|16.26|11.00|36.44|
> |||||||
> **Sign2GPT**|34.80|24.00|17.27|12.96|41.12|
> **Sign2GPT(w/ PGP)**|**41.75**|**28.73**|**20.60**|**15.40**|**42.36**|
> |||||||
> **Sign($Z$)2GPT(w/PGP)**|32.73|20.52|13.75|9.73|33.39|

---

> > ### Author Response · Authors · 2023-11-16
> > **Response to Reviewer hMSD (2/2)**
> >
> > **W4: I cannot find an ablation study on the pseudo-gloss pretraining. What is the performance if removing it?**
> >
> > The impact of the pseudo-gloss pretraining was shown in Tables 2 and 3 in the paper, but as you have pointed out the notation may have been confusing. The updated tables above should now clearly show the impact of the pretraining.
> >
> > **W5: In Table 4, removing sinusoidal positional encoding leads to the best performance. Then what poistional encoding is used by default?**
> >
> > Table 4 indicates that the inclusion of sinusoidal positional encoding yields superior performance. It's important to clarify that the ablation study focuses not on positional encoding at the start of the sign encoder but rather after the sign features $Z$. The improved performance can be attributed to the model's ability, during pretraining, to align sign features with pseudo-gloss prototypes if present in the sequence. Consequently, this leads to the learned features in pretraining discarding positional information in the output representation $Z$. The reintroduction of sinusoidal positional encoding allows the LLM to incorporate positional information in $Z$ for sign translation.
> >
> > **W6: The paper uses a new spatial backbone which is under-explored in previous sign language papers. The authors need to better motivate it. For example, is it better than other vision transformers, or is it better than widely adopted 2D/3D CNNs?**
> >
> > Our motivation for using Dino-V2 pretrained ViT model is discussed in our paper. We want a robust feature network that could be adapted with LoRA. Fully fine-tuning spatial models in previous approaches often involves a significant number of parameters that pose memory and computational challenges during training. With the adaptation of the ViT model, we are able to reduce the number of trainable parameters to under three hundred thousand, compared to the previous state-of-the-art approach GFSLT [2] which uses ResNet18 with around 11 million trainable parameters. We present ablation results below on the CSL-Daily using our approach without Pseudo-Gloss Pretraining to demonstrate the difference in performance by replacing our spatial model with a ResNet18 that has been trained on ImageNet. We find that using our spatial backbone has small improvements compared to using the conventional ResNet18 model.
> >
> > | Spatial Backbone | BLEU4 | ROUGE |
> > |-|-|-|
> > | ResNet18 | 12.28 |38.31|
> > | DinoV2 (ViT-S/14)  |**12.96**| **41.12**|
> >
> > **W7: In figure 2, how to fuse the outputs of adapted masked attention (solid lines) and zero-gated cross attention (dashed lines)? Besides, what is adapted masked attention? I didn't see a clear definition of it.**
> >
> > Adapted masked attention is the pretrained masked multi-head attention with LoRA as the trainable weights. The outputs of adapted masked attention and zero-gated cross attention are fused with summation since they are both of the same dimensions.
> >
> > The additional analyses, comparisons, and updates, will be incorporated into the revised paper, once the reviewer finds these inclusions satisfactory
> >
> > [1] Conditional sentence generation and cross-modal reranking for sign language translation. IEEE Transactions on Multimedia 2021
> >
> > [2] Gloss-free sign language translation: Improving from visual-language pretraining. CVPR 2023

---

> > > ### Comment · Reviewer_hMSD · 2023-11-23
> > >
> > > Thanks for the authors' rebuttal. My concerns are well addressed. The major issue is that the core idea of using pretrained model for SLT has already been studied, but the proposed new techniques, e.g., pseudo-gloss pretraining and using of LLM, can compensate the weakness. Thus, I tend to accept the paper but I am also open to discuss with other reviewers.

---

### Official Review · Reviewer_aj4n · 2023-10-30

**Soundness:** 4 excellent
**Presentation:** 3 good
**Contribution:** 3 good
**Rating:** 6
**Confidence:** 4

**Summary:**

This paper aims to improve gloss-free sign language translation by exploring pretrained vison and language models, i.e., ViT and GPT models. The authors use pretrained ViT models to extract spatial features from sign frames and use pretrained GPT models to perform the translation. They design a sign encoder and a zero-gated cross-attention module to bridge these two models and pretrain the encoder with a pseudo-gloss pretraining strategy. On Phoenix14T and CSL-Daily, this method achieves new SOTA in the gloss-free setting.

**Strengths:**

1) Introducing a simple method leveraging pretrained vision and language models to improve SLT: SOTA performance
2) Proposing a pretraining strategy based on pseudo glosses that induce meaningful sign representations

**Weaknesses:**

1) While achieving good performance, the pretrained models are significantly larger than previous approaches, raising concerns about fair comparison and what helps translation. Relevant ablation is missing.
2) Analysis regarding the sign encoder and pseudo-gloss pretraining is insufficient.
3) The used SLT benchmarks are somehow artificial albeit popular.
4) Some details are confusing.

**Questions:**

1) It's great that the proposed method outperforms GFSLT-VLP. However, GFSLT-VLP is based on MBart with ~600M parameters, and this study adopts XGLM with ~1.7B parameters, making the fairness of the comparison questionable. It's unclear whether the improvements are really from the proposed modeling and pretraining strategy. Could you please add further ablations regarding the size of GPT models?
2) The sign encoder and pseudo-gloss pretraining take a crucial role in the proposed method, but analysis and ablation regarding them are insufficient.
  - Do we need the sign encoder? What if dropping it?
  - Apart from the visual analysis, what about the top-N prediction accuracy and recall?
3) Please also add gloss-based SOTA systems in Tables 2 and 3 so that readers can understand the gap.
4) In Eq (1), you mentioned that V represents keys from textual features while K originates from sign features. Shouldn't they both come from sign features?
5) How did you set the rank in LoRA?
6) While Phoenix14T and CSL-Daily are popular, they are less significant due to limited vocabulary and data size. Please consider adding results for DGS3-T (Zhang et al., 2023).


After Response:

Thanks for the new results which address part of my concerns. I increased my score for this. Still, I believe the study of gloss-free approach should be performed on more realistic datasets, such as DGS3-T and WMT-SLT.

---

> ### Author Response · Authors · 2023-11-16
> **Response to Reviewer aj4n**
>
> Thanks for the review, and your insightful comments.
>
> **Q1: It's great that the proposed method outperforms GFSLT-VLP. However, GFSLT-VLP is based on MBart with ~600M parameters, and this study adopts XGLM with ~1.7B parameters, making the fairness of the comparison questionable. It's unclear whether the improvements are really from the proposed modeling and pretraining strategy. Could you please add further ablations regarding the size of GPT models?**
>
> While we acknowledge the substantial difference in parameter count, a key strength of our approach lies in the frozen pretrained language and vision model, with weight updates facilitated through adapters. This results in a total trainable parameter count of under 17 million, significantly less than GFSLT-VLP's ~600M parameters.
>
> To address the concern about the total parameter count, we conducted an ablation study using a smaller 564M parameter XGLM model. The results, presented in the tables below, reveal a marginal performance reduction on the test set of the Phoenix14T dataset.
> Even with similar total parameter counts, these results outperform GFSLT-VLP.
>
> |   | BLEU4 | ROUGE |
> |-----------|-----------|-----------|
> | GFSLT-VLP | 21.44  |42.49|
> | Sign2GPT (XGLM-564M) | 22.29  |48.21|
> | Sign2GPT (XGLM-1.7B) |  **22.52**|**48.90**|
>
> **Q2: The sign encoder and pseudo-gloss pretraining take a crucial role in the proposed method, but analysis and ablation regarding them are insufficient.
> Do we need the sign encoder? What if dropping it?
> Apart from the visual analysis, what about the top-N prediction accuracy and recall?**
>
> Thank you for your suggestion about quantitivate analysis for pretraining we will also include the below table of the precision, recall and F1-score for Phoenix14T and CSL-Daily pseudo-gloss.
>
> |  | Precision | Recall | F1-Score |
> |-----------|-----------|-----------|-----------|
> Phoenix14T  | 0.52   |  0.39 |  0.44   |
> CSL-Daily      |0.38 |   0.34  |   0.36 |
>
> To address the question about needing a sign encoder we have conducted the ablation study showing the pretraining precision, recall and F1-scores on the extracted pseudo-glosses shown in the table below. The role of the sign encoder forms an important part of the model as it learns temporal features which is reflected by the significant improvement in F1-score.
> |  | Precision | Recall | F1-Score |
> |-----------|-----------|-----------|-----------|
> No Sign Encoder  | 0.50    |   0.25   |   0.33   |
> Sign Encoder      |**0.52** |  **0.39** | **0.44** |
>
> **Q3: Please also add gloss-based SOTA systems in Tables 2 and 3 so that readers can understand the gap.**
>
> Thank you for your feedback, we will be updating the paper by adding gloss-based SOTA systems in the relevant tables.
>
> **Q4: In Eq (1), you mentioned that V represents keys from textual features while K originates from sign features. Shouldn't they both come from sign features?**
>
> Thanks for pointing that out, you are correct that V and K both originate from sign features, we will clarify this in the manuscript.
>
> **Q5: How did you set the rank in LoRA?**
>
> During our experiments, we set the LoRA rank and alpha values both as 4.
>
> **Q6: While Phoenix14T and CSL-Daily are popular, they are less significant due to limited vocabulary and data size. Please consider adding results for DGS3-T (Zhang et al., 2023).**
>
> Current benchmarks for state-of-the-art gloss-free results primarily focus on Phoenix14T and CSL-Daily. We appreciate the suggestion to include results for DGS3-T. However, we encountered challenges in obtaining the dataset as the GitHub instructions for downloading are currently non-functional.
>
> To address this, we have contacted the authors of DGS3-T to explore the feasibility of obtaining the dataset. If successful, we intend to include the results as baselines for gloss-free sign translation, in the appendices in the CRC.
>
> The additional analyses, comparisons, and updates, will be incorporated into the revised paper, once the reviewer finds these inclusions satisfactory.

---

### Official Review · Reviewer_DHgv · 2023-10-30

**Soundness:** 3 good
**Presentation:** 3 good
**Contribution:** 3 good
**Rating:** 6
**Confidence:** 4

**Summary:**

The paper proposes an automatic sign language translation based on large language models.
The amount of training data for sign language is limited, but the authors present an idea to leverage the large-scale resources from spoken language.
The proposed framework utilises large-scale vision and language models with lightweight adapters.
In particular, the method leverages a fronzen GPT model for translation.
The method is gloss-free, so that large-scale data can be used without gloss-level annotations for supervised learning.
Instead, the authors propose a novel encoder pretraining strategy based on pseudo-gloss that can be extracted from natural language sentences.
The authors experiment on the popular PHOENIX and CSL-Daily datasets, on which they demonstrate state-of-the-art performance compared to all baselines.

**Strengths:**

- The use of LLM for sign language recognition is novel and effective.
- The gloss-free framework using pseudo-gloss mitigates the challenging supervision problem in sign language recognition.
- The results are state-of-the-art, and the ablations in Tables 2 and 3 show that the proposed pre-training helps to improve performance.
- The writing is generally clear.

**Weaknesses:**

- The method is mostly based on existing models such as GPT, Dino-V2 and LoRA, so there is not much novelty from the architectural standpoint.
- There is no ablations to demonstrate if the choice to use parts-of-speech tagging effective. How does it compare to using the words as tokens? Can the downstream translation learn to generate meaningful words that are missing in the parts-of-speech tagging?

**Questions:**

Please see the last point of weaknesses.

---

> ### Author Response · Authors · 2023-11-16
> **Response to Reviewer DHgv**
>
> Thanks for the review, and your insightful comments.
>
> **W1: The method is mostly based on existing models such as GPT, Dino-V2 and LoRA, so there is not much novelty from the architectural standpoint.**
>
> While we have used existing models, we introduce a novel technique and model architecture to break the reliance on gloss which is something that everyone working in sign translation would be interested in.
> Moreover, our paper highlights technical considerations inherent to sign language translation, including the handling of numerous frames leading to memory constraints and the challenge posed by limited dataset size which we address with our architectural design of using frozen pretrained models with adapters.
> We also introduce a pretraining strategy specific to our architecture, enabling the learning of pseudo-gloss localization which serves as a crucial representation for the LLM.
>
> **W2: There is no ablations to demonstrate if the choice to use parts-of-speech tagging effective. How does it compare to using the words as tokens? Can the downstream translation learn to generate meaningful words that are missing in the parts-of-speech tagging?**
>
> The downstream translation effectively learns to replace the missing words as demonstrated by our qualitative translation examples in Tables 5 and 6. This is a key benefit of a strong language model.
>
> Regarding the ablation of parts-of-speech (POS) tagging, we deliberately excluded certain parts of speech as we know they are not represented in sign, e.g. connected words such as 'and' and articles such as 'the' lack a functional sign equivalent. In the table below, we illustrate the impact of utilizing POS on precision, recall, and F1-score for our pretraining results on Phoenix14T with a threshold of 0.2. Notably, when all words are used as tokens, recall significantly decreases from 0.39 to 0.28. This result validates our assertion that not all words have corresponding signs in sign language.
> As outlined in the paper, this pretraining strategy opens avenues for future research, particularly in areas like sign spotting.
>
> | Tokens | Precision | Recall | F1-Score |
> |-----------|-----------|-----------|-----------|
> | All Words | **0.55** |0.28|0.37|
> | Pseudo-Glosses | 0.52| **0.39**|**0.44**|
>
> The additional analyses, comparisons, and updates, will be incorporated into the revised paper, once the reviewer finds these inclusions satisfactory.

---

> ### Comment · Reviewer_DHgv · 2023-11-23
>
> Thanks for the additional information. I maintain the view that I learn towards acceptance.

---

### Official Review · Reviewer_3mSZ · 2023-11-03

**Soundness:** 2 fair
**Presentation:** 2 fair
**Contribution:** 2 fair
**Rating:** 5
**Confidence:** 5

**Summary:**

This paper aims to leverage the large-scale pretrained vision and language models via lightweight adapters for gloss-free sign language translation. Besides, it also proposes a pretraining strategy which make the framework aware of important text information. The experiments are conducted on two benchmarks to validate the effectiveness of the proposed method.

**Strengths:**

Leveraging large-scale pretrained model for SLT is sound.

The paper is well-written and well-organized.

The overall performance seems promising.

**Weaknesses:**

The introduction part seems inconsistent with the title. The introduction mentions the both large-scale vision and language model, while the title only mentions the language one. What do the authors want to emphasize?

The pseudo-gloss pretraining technique shares the similar core idea with CSGCR. The authors should discuss the difference.

Utilization of pretrained model is not new. The author should cite the following work and discuss the difference with it.
Chen Y, Wei F, Sun X, et al. A simple multi-modality transfer learning baseline for sign language translation[C]//Proceedings of the IEEE/CVF Conference on Computer Vision and Pattern Recognition. 2022: 5120-5130.

What are the performance gains derived from the utilization of large-scale vision and language models? The ablation part should demonstrate it.

**Questions:**

See the Weakness section.

---

> ### Author Response · Authors · 2023-11-16
> **Response to Reviewer 3mSZ (1/2)**
>
> Thanks for the review, and your insightful comments.
>
> **W1: The introduction part seems inconsistent with the title. The introduction mentions the both large-scale vision and language model, while the title only mentions the language one. What do the authors want to emphasize?**
>
> We appreciate the reviewer's careful examination of the introduction and title. Our emphasis on 'leveraging large language models' in the title is intentional. Our goal is to create a sign model/representation that is passed to / leveraged by the GPT model for sign language translation, hence the name Sign2GPT.
> The 'Sign' component that is leveraged by the GPT model includes the use of a large vision model and our novel pretraining strategy.
> However, we are more than happy to change the title to address this issue.
>
> **W2: The pseudo-gloss pretraining technique shares the similar core idea with CSGCR. The authors should discuss the difference.**
>
> We appreciate the reviewer's insights into CSGCR [1] and observe these distinctions between our proposed pseudo-gloss pretraining technique and CSGCR:
> - *Training Approach* - CSGCR employs separate transformer encoders for each vocabulary item, necessitating positive and negative samples during training. In contrast, our method utilizes a single transformer encoder for all pseudo-glosses, ensuring scalability for larger vocabularies.
> - *Localization Capabilities* - Our pretraining strategy exhibits inherent explicit localization capabilities, as presented in Figures 4, 5, and 6.
> - *Translation Process* - Unlike CSGCR's two-step process involving Video2Text (Word Existence Verification) and then Text2Text (Sentence Generation), our approach directly utilizes the pretrained model for Video2Text using the GPT model.
>
> We will update the paper to discuss these differences in the related work.
>
> **W3: Utilization of pretrained model is not new. The author should cite the following work [2] and discuss the difference with it.**
>
> Thanks for highlighting the reference. We will incorporate a discussion of its relevance in the related work section. To outline the distinctions, Chen et al. [2] focus on utilizing manually annotated sign-ordered gloss supervision, relying on Gloss2Text for progressive pretraining on the encoder-decoder language model. In contrast, our approach has a significant advantage of not depending on manually annotated glosses. Instead, we concentrate on developing a method for gloss-free sign representation, leveraging a decoder-only language model without the need for progressive pretraining of the language model.

---

> > ### Author Response · Authors · 2023-11-16
> > **Response to Reviewer 3mSZ (2/2)**
> >
> > **W4: What are the performance gains derived from the utilization of large-scale vision and language models? The ablation part should demonstrate it.**
> >
> > In the table below, we present the results of an ablation experiment on CSL-Daily, comparing the performance of a ResNet18 backbone used in GFSLT [3] with our Adapted DinoV2 approach within our Sign2GPT architecture. Note this ablation study does not involve pseudo-gloss pretraining. The findings indicate that the model trained with DinoV2 pretraining, incorporating adapters, exhibits slightly better performance than the ResNet18 backbone utilized in the previous state-of-the-art GFSLT approach.
> >
> > | Spatial Backbone | BLEU4 | ROUGE |
> > |-|-|-|
> > | ResNet18 | 12.28 |38.31|
> > | DinoV2 (ViT-S/14)  |**12.96**| **41.12**|
> >
> > A noteworthy advantage of our approach lies in the application of LoRA instead of fine-tuning all parameters. Consequently, the proposed backbone features under three hundred thousand trainable parameters, a significant reduction compared to the 11 million trainable parameters of the ResNet18 model.
> >
> > Additionally, we commit to updating Tables 2 and 3 in the paper to include GFSLT approach without their vision-language pretraining. The tables below demonstrate that our architecture even without pseudo-gloss pretraining (w/o PGP) can outperform it. This sufficiently demonstrates that our architectural design also shows performance improvements compared to the previous attempts at the utilization of pretrained language models.
> >
> > *Phoenix14T Translation Results:*
> > Method|BLEU1|BLEU2|BLEU3|BLEU4|ROUGE|
> > |-|-|-|-|-|-|
> > |*No Sign Prior Pretraining*| |||||
> > GFSLT|41.39|31.00|24.20|**19.66**|40.93|
> > Sign2GPT (w/o PGP)|**45.43**|**32.03**|**24.23**|19.42|**45.23**|
> > |*With Sign Prior Pretraining*| |||||
> > GFSLT-VLP|43.71|33.18|26.11|21.44|42.49|
> > Sign2GPT(w/ PGP)|**49.54**|**35.96**|**28.83**|**22.52**|**48.90**|
> >
> > *CSL-Daily Translation Results:*
> > Method|BLEU1|BLEU2|BLEU3|BLEU4|ROUGE|
> > |-|-|-|-|-|-|
> > |*No Sign Prior Pretraining*| |||||
> > GFSLT|**37.69**|23.28|14.93|9.88|35.16|
> > Sign2GPT (w/o PGP)|34.80|**24.00**|**17.27**|**12.96**|**41.12**|
> > |*With Sign Prior Pretraining*| |||||
> > GFSLT-VLP|39.37|24.93|16.26|11.00|36.44|
> > Sign2GPT (w/ PGP)|**41.75**|**28.73**|**20.60**|**15.40**|**42.36**|
> >
> > The additional analyses, comparisons, and updates, will be incorporated into the revised paper, once the reviewer finds these inclusions satisfactory.
> >
> > [1] Conditional sentence generation and cross-modal reranking for sign language translation. IEEE Transactions on Multimedia 2021
> >
> > [2] A simple multi-modality transfer learning baseline for sign language translation. CVPR 2022
> >
> > [3] Gloss-free sign language translation: Improving from visual-language pretraining. CVPR 2023

---

### Author Response · Authors · 2023-11-21

Thanks everyone for their reviews. It appears that all reviewers commended our approach to achieving **SOTA performance in Gloss-Free Sign Language Translation** (*"overall performance seems promising"*, *"novel and effective"*, *"It is good to see that current large language model can be helpful for sign language understanding."*). They think that our **pseudo-gloss pretraining** *"induce meaningful sign representations"* and *"is novel, which can inspire future works"*.

However, while achieving state-of-the-art results, the main concerns of reviewers were (1) *differences to previous approaches*; (2) *the effectiveness of the large-scale vision and language models*; (3) *further analysis of our proposed pseudo-gloss pretraining*.
We believe we have addressed all of these concerns as reflected by the updated manuscript and response to reviewers.

(1) We have highlighted that our approach is *advantageous over previous approaches* as it **does not require manually annotated sign-ordered glosses**. Our **novel pretraining strategy** is independent of the decoder-only LLM and makes use of **lightweight adapters** to leverage powerful vision and language models for Sign Language Translation. The pseudo-gloss pretraining also has inherent **automatic localization capabilities** which can aid future sign language research.

(2) We have provided **new ablation studies** that verify the effectiveness of our proposed architecture by making more direct comparisons with previous approaches for large-scale vision (Table 7) and language models (Table 11).

(3) We have also addressed concerns about our proposed pseudo-gloss pretraining having only qualitative results by including **new quantitive results** on precision, recall, and F1-score (Table 8). The inclusion of **ablation of pseudo-gloss pretraining** also gives justification for using the selected pseudo-glosses (Table 9) and sign encoder (Table 10).

We believe that addressing the challenging problem of gloss-free translation has wide-reaching interest to the ICLR community. In addition, breaking the reliance on gloss annotation is absolutely key to making progress in sign language translation. Gloss-free sign language translation is an active area of research and our work demonstrates an effective approach to learning meaningful sign representations that can be directly leveraged by large language models to achieve SOTA results.

---

### Meta-Review · Area_Chair_BTjy · 2023-12-05

**Metareview:**

The paper aims to improve sign language translation in a gloss-free setup with pretrained vision and language models. It uses adapters to finetune the pretrained models and proposes a new pseudo-gloss pretraining strategy to learn sign representation. Experimental results on two benchmarks demonstrate the effectiveness of the proposed method.

The initial concerns were lacking ablations and the idea of using pretrained models has been explored in previous work. The rebuttal provided sufficient results to address the concerns on the ablations, and clarified the differences compared to prior work. Three reviewers out of four recommended acceptance.
Considering that the paper is well written and the proposed pseudo-gloss pretraining is new with supported experiments, the AC agrees with the majority of the reviewers and recommends acceptance.
The authors should include the new ablations and comparison with prior work in the final version.

**Justification For Why Not Higher Score:**

The proposed method is not significantly novel.

**Justification For Why Not Lower Score:**

The work addresses a practical challenging gloss-free sign language translation setting. The proposed method is solid with good empirical results.

---

### Decision · Program_Chairs · 2024-01-16

Accept (poster)